# Surgical Excision of Unusual Sacked Neck and Mediastinum Abscess of Odontogenic Origin

**DOI:** 10.3390/antibiotics11121757

**Published:** 2022-12-05

**Authors:** Andrea Colizza, Giovanni D’Erme, Andrea Ciofalo, Giacomo D’Angeli, Francesca Romana Federici, Carlotta Galli, Marco De Vincentiis, Massimo Galli

**Affiliations:** 1Department of Sense Organs, Sapienza University of Rome, 00186 Roma, Italy; 2Department of Oral and Maxillofacial Sciences, Sapienza University of Rome, 00161 Roma, Italy; 3Department of Medical Biotechnology, University of Siena, 53100 Siena, Italy

**Keywords:** abscess, odontogenic infection, surgical drain, tracheostomy

## Abstract

The most common cause of neck infections is odontogenic abscesses that can often be life-threatening and require a surgical drain associated with antibiotic therapy. We present a case of the surgical management of an odontogenic sack-shaped and walled abscess arising from elements 3.6, 3.7 and 3.8 that reached the laterocervical spaces and anterior mediastinum in a 28-year-old healthy woman. Typical signs and symptoms of cervical complications of dental origin are fever, a neck mass, lymphadenopathy, trismus and odynophagia. The gold standard treatment in these situations is a multidisciplinary approach involving an oral surgeon, ENT specialist and thoracic surgeon to drain the infected material. To the best of our knowledge, this is the first described case report of a dental abscess enclosed in a sack in the deep space of the neck and in the anterior space of the mediastinum.

## 1. Introduction

Odontogenic abscesses are the most common cause of neck infections [1]. The removal of the infection starting from dental and periodontal structures, with an appropriate antibiotic therapy, usually leads to complete healing. The pathogenesis is polymicrobial, consisting of various facultative and strict anaerobes. The dominant microorganism are strictly anaerobic Gram-negative bacteria and Gram-positive cocci [1].

There are conflicting reports in the literature regarding drug therapy for the treatment of odontogenic abscesses. Empirically, when taking into account the multiplicity of the oral cavity microbiota [2], we tend to administer an antibiotic combination comprising a broad-spectrum antibiotic and another one more suitable for Gram-negative and anaerobic bacteria.

On the contrary, the literature unanimously agrees on the necessity of the surgical drainage of the infection.

If it is not treated early enough, the infection could spread to nearby structures, such as soft tissues and the skin of the cervicofacial area, maxillary sinus, the superficial and deep space of the neck and even up to the mediastinum and thorax in more severe cases. These conditions can be life-threatening and require the removal of the primary source of infection through a surgical incision and the drainage of the abscess in deep neck spaces. Moreover, in the case of a reduction in the upper airway tract, a tracheostomy is required, [3,4]. The surgery of the neck, even if in emergency cases, requires an aesthetic incision with respect to the Langer lines [5,6].

## 2. Case Report

Herein, we present the case of a 28-year-old woman admitted to our emergency department presenting with left submandibular swelling that had arisen five days earlier, and was associated with a fever, progressive dysphagia, trismus and dyspnea. The ENT examination revealed a painful left submandibular swelling covered with hyperemic and warm skin. The endoscopy of the upper airway tract showed swelling on the left side of the oropharynx and hypopharynx, and a mobility impairment of the left vocal fold. The head–neck computed tomography (CT) scan performed with and without a contrast agent showed a voluminous mass involving the left submandibular region, perimandibular soft tissues, masseter muscle, tonsillar region and the parapharyngeal space, resulting in the aerial column deviating to the right. The swelling caused the compression of the internal jugular vein and extended up to the anterior mediastinum, behind the sternal manubrium (Figure 1A,B).

The CT showed that the probable origin of the abscess was an odontogenic infection coming from the inferior left molars (Figure 1C).

The patient was transferred at the ENT ward and was given antibiotics (2.2 g IV (intravenous) amoxicillin every 8 h and 600 mg IV clindamycin every 6 h) and corticosteroid therapy (4 mg IV betamethasone every 12 h). We planned a double approach: an intraoral access—removing the teeth that probably caused the infection—and an extraoral one—performing a surgical drainage of the abscess in the neck. In detail, the oral surgeon realized a full thickness flap through a buccal and lingual incision of the oral soft tissue from element 3.4 up to element 3.8; he then performed the avulsion of elements 3.6 and 3.7, and, finally, through osteotomy, the extraction of element 3.8. Subsequently, the ENT specialist performed a left Paul-Andre neck incision to open the submandibular lodge and the laterocervical space. After the drainage of the deep space of the neck, we found a membrane containing the purulent material originating from the tooth roots of element 3.8, which stretched up from the submandibular and laterocervical region to the mediastinum, measuring 13 cm (Figure 2A,B).

Once the membrane was removed, the thoracic surgeon performed an exploration of the anterior mediastinum, which confirmed the patient was free from further infected material. Then, a Penrose drain was positioned in the oral cavity, one Redon drain in the submandibular lodge, one in the laterocervical space and another one in the upper anterior mediastinum. At the end, the ENT surgeon performed a tracheostomy to protect the airways.

Wound dressing changes were daily performed during the postoperative recovery. The tracheostomy was closed 10 days after the surgery, and the patient had a CT scan performed 15 days after the surgery (Figure 3A,B), which revelated the complete healing of the tissues. The patient was discharged after 19 days of hospitalization.

## 3. Discussion

Odontogenic abscesses often derive from dental plaque, caries and pulp necrosis, and in 90% of cases, the inferior molars are the focus of the infection [7].

The most frequent bacteria responsible for caries and pulp necrosis are facultatively anaerobic, Gram-positive coccus, such as *Streptococcus* spp. (S. Mitis and S. Mutans) and Staphylococcus Aureus, which reach the periapical region through the root canal system [8].

Another common cause of an odontogenic abscess is untreated chronic periodontitis, leading to exacerbation. Putative pathogens include Gram-negative (G-) anaerobic bacteria, spirochetes and even viruses [9].

The plausible different nature of this kind of infection justifies the use of double-drug therapy comprising a broad-spectrum antibiotic and another one more specific for anaerobic G- microorganisms.

The long sack-shaped membrane we showed likely arose from an untreated odontogenic abscess. In this case, it was important to consider a differential diagnosis of a congenital cyst. However, this diagnostic hypothesis was ruled out on the basis of the CT images, which clearly highlighted the odontogenic origin of the infection.

The typical signs and symptoms of cervical complications from dental origin are fever, neck mass, lymphadenopathy, trismus and odynophagia. Furthermore, systemic complications can occasionally occur, such as Lemierre’s syndrome, which is a rare and potentially life-threatening metastatic infection, typically characterized by the suppurative thrombophlebitis of the internal jugular vein and the subsequent sprawl—through the bloodstream—in the lungs and various other organs, such as the liver and brain [10].

In the literature, there are no clear guidelines for the correct surgical management of odontogenic deep neck space infections; generally, these involve airway impairments, sepsis, descending infections, abscesses larger than 3 cm and the involvement of deep neck spaces [11]. The main purpose of surgery is to eliminate the cause of infection and drain the purulent material. For this reason, a specialist team comprising an oral surgeon, ENT specialist and, when necessary, a thoracic surgeon is mandatory. The correct surgical management is planned on the clinical and radiological diffusion of the abscess. Infections that spread from the mandible to the submandibular, parapharyngeal and laterocervical space are approached through an external Paul-Andre approach [12]. The neck dissection aims to identify the sternocleidomastoid (SCM) muscle, the superficial and middle cervical fascia and the neurovascular bundle. In the most complicated situations, the infectious process can reach the mediastinum through the perivisceral space. Descending mediastinitis was classified by Endo et al. [13], according to the computed tomography findings (Figure 4).

In these situations, surgery is a complex procedure, and generally requires an extended cervicotomy in order to drain the abscess. The mortality rate for this infection, in recent years, has decreased from approximately 50% to 18% [14,15]. The most important factors for mortality are the degree of infection and the patient’s underlying diseases and comorbidities (such as diabetes, HIV infection, etc.). The causes of death are multiple, and the most frequent ones are septic shock, respiratory insufficiency, and gastrointestinal hemorrhages [14]. The main challenge for ENT specialists is preserving the airway tract. Adley et al. [16] analyzed the advantage of performing an early tracheostomy rather than intubating, in order to reduce the sedation, mechanical ventilation, and faster discharge of the patient. To the best of our knowledge, this was the first described case report of a dental abscess enclosed in a 13 cm long sack-shaped membrane in the deep space of the neck and in the anterior space of the mediastinum.

## 4. Limits of the Study

In this case report, we showed an unusual form of an odontogenic abscess. Due to the presence of purulent material, the principal diagnosis was an infected cyst of the neck. Unfortunately, we did not perform a histological examination because, in our humble opinion, during the surgery, it was implied that the sack-shaped formation arose from an inflammatory process of the broken inferior molar.

## Figures and Tables

**Figure 1 antibiotics-11-01757-f001:**
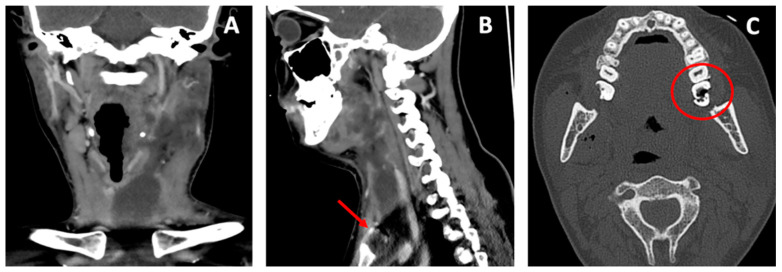
Computed tomography scan of the abscess affecting the deep space of the neck and the anterior mediastinum in coronal view (**A**). The red arrow in sagittal view shows the abscess behind the sternal manubrium (**B**). In the axial view the red circle shows the broken third inferior left molar (**C**).

**Figure 2 antibiotics-11-01757-f002:**
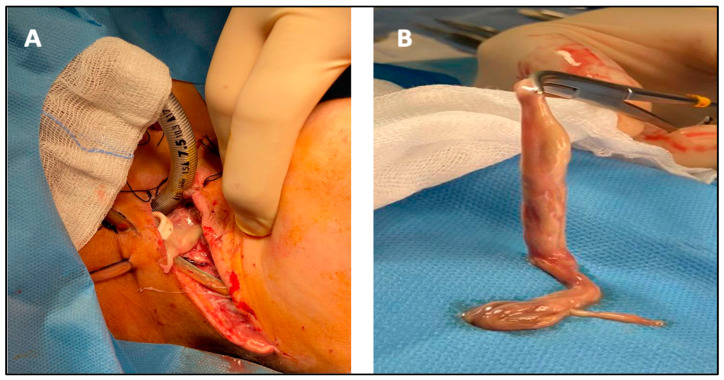
The wall of the abscess in the neck during the surgical drain (**A**) and after the removal (**B**).

**Figure 3 antibiotics-11-01757-f003:**
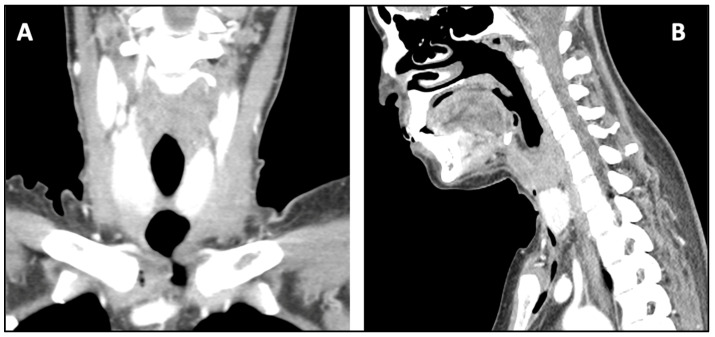
CT scan performed 15 days after surgical drain shows the healing in coronal view (**A**) and in sagittal view (**B**).

**Figure 4 antibiotics-11-01757-f004:**
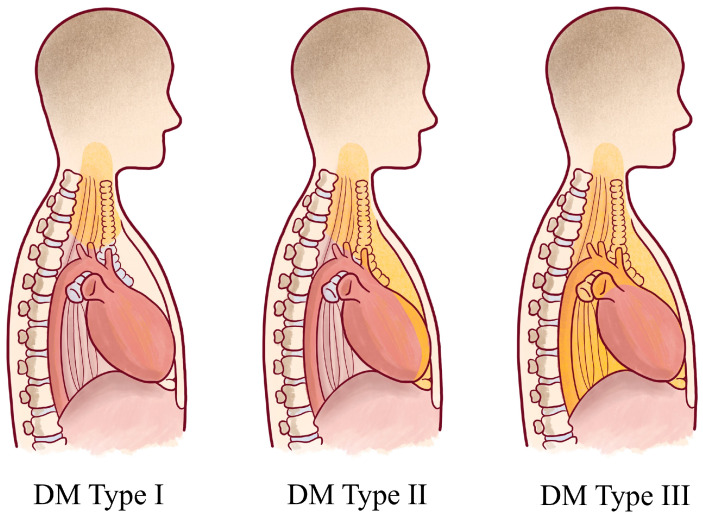
Descending mediastinitis (DM) classifications. The grey area indicates the extension of infections. Type I: DM is localized in the upper mediastinum above the tracheal bifurcation. Type II: DM extends to the lower anterior mediastinum. Type III: DM extends to the anterior and lower posterior mediastinum.

## Data Availability

Not applicable.

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
