# Peer review of "Surgical Excision of Unusual Sacked Neck and Mediastinum Abscess of Odontogenic Origin"

_antibiotics, 2022, doi:10.3390/antibiotics11121757_

Round 1

Reviewer 1 Report

The authors present a case report on surgical excavation of a cystic lesion ("sacked neck and mediastinum abscess") of odontogenic origin. The english language should be improved. The authors should discuss more sufficiently why they do not consider an infected (e.g. embryogenetic) cyst or wether this could be a differential diagnosis. The authors do not present histologic findings. Altogether, the case report could potentially be of interest, but further deliberations should be made.

Author Response

    Rome, November, 06, 2022

Manuscript: antibiotics-1965071 (Surgical excision of unusual sacked neck and mediastinum abscess of odontogenic origin).

Dear Reviewers,

We wish to thank You for reviewing our manuscript and for the valuable comments that helped us to clarify some relevant aspects that were missed or unclear in the first version of the paper. We have read the comments and made the changes to address comments and concerns. The manuscript has been reviewed by a native English speaker for grammar and language improvement.

We hope that the changes made in the revised manuscript and responses provided below have adequately addressed the comments and made this paper stronger.

REVIEWER #1

The authors present a case report on surgical excavation of a cystic lesion ("sacked neck and mediastinum abscess") of odontogenic origin. The english language should be improved.

We thank the reviewer for the careful review and for the comments. The manuscript has been reviewed by a native English speaker for grammar and language improvement.

The authors should discuss more sufficiently why they do not consider an infected (e.g. embryogenetic) cyst or wether this could be a differential diagnosis.

We share the comment of the reviewer. We didn’t consider among the differential diagnosis an infected embryogenetic cyst of neck space because the infected sack originated from the broken tooth roots of 3.8 dental element. We added in the manuscript another CT imagine that shows the dental origin of the abscess. In the discussion section we clarify the reasons for the exclusion of this diagnostic hypothesis.

The authors do not present histologic findings.

We share the comment of the reviewer but we didn’t performed the histologic examination on the sac.

Altogether, the case report could potentially be of interest, but further deliberations should be made.

We thank the reviewer. We performed the suggested correction and revisions in order to made more interesting the case.

We really appreciated your careful and thoughtful evaluation of our manuscript and hope that this revised version meets with your approval. We have tracked all changes using the “track changes” tool of Microsoft Word.

Thanks again for your interest in our work. We await your review of our revised manuscript.

Sincerely yours,

The Authors

Reviewer 2 Report

- Descending mediastinitis after an odontogenic neck infection is not novel

- The authors do not provide evidence for the odontogenic origin of the infection

- The nature of the "sac" is not revealed or discussed. Could this be an infected congenital cyst? It does not seem like a typical abscess wall either. If it is an infected congenital cyst, what did the purported odontogenic infection have to do with it?

- The references of descending mediastinitis is old (2006) and the high mortality rate (50%) may be an exaggeration in modern medicine. A more recent review suggest an average mortality rate of 18% (https://doi.org/10.1002/hed.24183)

Author Response

Manuscript: antibiotics-1965071 (Surgical excision of unusual sacked neck and mediastinum abscess of odontogenic origin).

Dear Reviewers,

We wish to thank You for reviewing our manuscript and for the valuable comments that helped us to clarify some relevant aspects that were missed or unclear in the first version of the paper. We have read the comments and made the changes to address comments and concerns. The manuscript has been reviewed by a native English speaker for grammar and language improvement.

We hope that the changes made in the revised manuscript and responses provided below have adequately addressed the comments and made this paper stronger.

REVIEWER #2

Descending mediastinitis after an odontogenic neck infection is not novel

- The authors do not provide evidence for the odontogenic origin of the infection

We thank the reviewer for the comment. The infected sack originated from the broken tooth roots of 3.8 dental element. We added in the manuscript another CT imagine that shows the dental origin of the abscess.

- The nature of the "sac" is not revealed or discussed. Could this be an infected congenital cyst? It does not seem like a typical abscess wall either. If it is an infected congenital cyst, what did the purported odontogenic infection have to do with it?

We share the comment of the reviewer. We didn’t consider among the differential diagnosis an infected embryogenetic cyst of neck space because the infected sack originated from the broken tooth roots of 3.8 dental element.. We added in the manuscript another CT imagine that shows the dental origin of the abscess. In the discussion section we clarify the reasons for the exclusion of this diagnostic hypothesis.

- The references of descending mediastinitis is old (2006) and the high mortality rate (50%) may be an exaggeration in modern medicine. A more recent review suggest an average mortality rate of 18% (https://doi.org/10.1002/hed.24183).

We thank the reviewer for the comment. We added the suggested references in the text.

We really appreciated your careful and thoughtful evaluation of our manuscript and hope that this revised version meets with your approval. We have tracked all changes using the “track changes” tool of Microsoft Word.

Thanks again for your interest in our work. We await your review of our revised manuscript.

Sincerely yours,

The Authors

Reviewer 3 Report

many thanks for the authors submission. This is a nicely and interesting submission. However some modifications are needed.

Introduction
This section goes straight to the point but underevaluate the need of aesthetic surgical approach in the neck.
At the line 38 the authors should should add this phrase "...Surgery of the neck, even if in emergency cases, requires an aesthetic incision with respect of the Langer lines"
Please cite the following
Yazdani Abyaneh MA, Griffith R, Falto-Aizpurua L, Nouri K. Famous lines in history: Langer lines. JAMA Dermatol. 2014 Oct;150(10):1087. doi: 10.1001/jamadermatol.2014.659. PMID: 25321654.
Aboh IV, Chisci G, Salini C, Gennaro P, Cascino F, Gabriele G, Iannetti G. Submandibular ossifying lipoma. J Craniofac Surg. 2015 May;26(3):973-4. doi: 10.1097/SCS.0000000000001489. PMID: 25933156.

Discussion
please revise the use of english as there are many errors "(i.e. line 103 ".. for this reason is mandatory..."

line 141 please remove the " ..".."

Author Response

Manuscript: antibiotics-1965071 (Surgical excision of unusual sacked neck and mediastinum abscess of odontogenic origin).

Dear Reviewers,

We wish to thank You for reviewing our manuscript and for the valuable comments that helped us to clarify some relevant aspects that were missed or unclear in the first version of the paper. We have read the comments and made the changes to address comments and concerns. The manuscript has been reviewed by a native English speaker for grammar and language improvement.

We hope that the changes made in the revised manuscript and responses provided below 

have adequately addressed the comments and made this paper stronger. 

REVIEWER #3

many thanks for the authors submission. This is a nicely and interesting submission. However some modifications are needed.

Introduction
This section goes straight to the point but underevaluate the need of aesthetic surgical approach in the neck.

At the line 38 the authors should should add this phrase "...Surgery of the neck, even if in emergency cases, requires an aesthetic incision with respect of the Langer lines"

Please cite the following Yazdani Abyaneh MA, Griffith R, Falto-Aizpurua L, Nouri K. Famous lines in history: Langer lines. JAMA Dermatol. 2014 Oct;150(10):1087. doi: 10.1001/jamadermatol.2014.659. PMID: 25321654.

Aboh IV, Chisci G, Salini C, Gennaro P, Cascino F, Gabriele G, Iannetti G. Submandibular ossifying lipoma. J Craniofac Surg. 2015 May;26(3):973-4. doi: 10.1097/SCS.0000000000001489. PMID: 25933156.

We thank the reviewer for the careful review and for the comments. We added in the manuscript the sentences and the suggested references in the introduction section.

Discussion
please revise the use of english as there are many errors "(i.e. line 103 ".. for this reason is mandatory..."

We thank the reviewer for the careful review. The manuscript has been reviewed by a native English speaker for grammar and language improvement. We corrected the sentence accordingly with your suggestion.

line 141 please remove the " ..".."

We corrected the sentence accordingly with your suggestion.

We really appreciated your careful and thoughtful evaluation of our manuscript and hope that this revised version meets with your approval. We have tracked all changes using the “track changes” tool of Microsoft Word.

Thanks again for your interest in our work. We await your review of our revised manuscript.

Sincerely yours,

The Authors

Round 2

Reviewer 1 Report

Although the authors improved the manuscript now, this reviewer must insist, that -by definition- abscesses are not located in an epithelial sac or membrane. The existence of such a sac leads to the definition of a "cystic formation".

Such thing would be by far more obvious. To this reviewers regret, histologic evaluation is missing and -although the tooth 38 was broken in half-  it might likewise not have been the source for the "sac". 

The paper could be published after all, if the authors could provide histologic findings to demonstrate odontogenic source in a superior way.

Author Response

Dear reviewers, 

We hope that the changes made in the revised manuscript and responses provided below have adequately addressed the comments and made this paper stronger.

The existence of such a sac leads to the definition of a "cystic formation". We share the comment of the reviewer but we didn't perform histology so we didn't define in as a cyst and we preferred to continue calling it abscess, based on the presence of purulent material. Besides, contrary to the phlegmon, the abscess si bounded by a wall, just like in this case.

We have also added a paragraph to explain the limit of the study and the reason why we didn't perform histology.

Although there is no histological documentation, we think that this case report is still interesting and it should be present in the literature. 

Thank you for your advice.

Reviewer 2 Report

The authors have adequately addressed the reviewers' comments.

Author Response

Dear reviewers,

We hope that the changes made in the revised manuscript and responses provided below have adequately addressed the comments and made this paper stronger.

Thank you for your advices.

Reviewer 3 Report

Accept

Author Response

Dear reviewers,

We hope that the changes made in the revised manuscript and responses provided below have adequately addressed the comments and made this paper stronger.

Thank you for your advice.